# ATM-deficient murine thymic T-cell lymphoblastic lymphomas are PTEN-deficient and require AKT signaling for survival

Joseph B. An[1]⊕, Karen S. Hathcock[1]⊕, Seth M. Steinberg[2], Hyoyoung M. Choo-Wosoba[2], Richard J. Hodes[1,3]*

1 National Cancer Institute (NCI), National Institutes of Health (NIH), Experimental Immunology Branch, Bethesda, MD, United States of America, 2 Biostatistics and Data Management Section, Center for Cancer Research, NCI, NIH, Bethesda, Maryland, United States of America, 3 National Institute on Aging (NIA), NIH, Bethesda, MD, United States of America

⊕ These authors contributed equally to this work.
* hodesr@31.nia.nih.gov

**Data Availability Statement:** All relevant flow data, raw images, and Excel files for this study are publicly available from the figshare repository (https://doi.org/10.6084/m9.figshare.27129045.

## Abstract

Mice deficient in the ataxia telangiectasia mutated (ATM) kinase have impaired responses to genotoxic and oxidative stressors, predisposing them to develop thymic T-cell lympho-blastic lymphomas (T-LBL) resembling human T-cell acute lymphoblastic leukemias (T-ALL). A previous study identified genomic deletions of the gene encoding PTEN, a negative regulator of PI3K/AKT/mTOR signaling, in a subset of murine ATM-deficient (ATMKO) thymic T-LBLs; however, the frequency and consequences of these deletions were not defined. The present study demonstrates that the majority of established cultures of ATMKO T-LBLs isolated from ATMKO thymi have a variety of genomic *Pten* alterations and fail to express functional PTEN protein. In addition, all T-LBLs demonstrate constitutive expression of pAKT, indicating the presence of activated AKT signaling, and are sensitive to treatment with the pan-AKT inhibitor MK-2206, suggesting that these lymphomas are dependent on pAKT signaling for their survival. Lastly, ATM-deficiency itself does not cause loss of PTEN or dysregulated AKT signaling, as ATM-deficient non-malignant thymocytes express wild-type levels of PTEN and lack detectable pAKT. This study demonstrates for the first time that the majority of ATM-deficient thymic T-LBLs lose PTEN expression and all depend on AKT signaling for survival, suggesting their potential use as an animal model for PI3K/AKT/MTOR pathway dysfunction in human T-ALL.

## Introduction

Ataxia telangiectasia mutated (ATM) is a serine/threonine kinase that responds to DNA double-stranded breaks and oxidative stressors [1,2] by eliciting cell cycle arrest, DNA repair, or p53-dependent apoptosis [3,4]. ATM is a tumor suppressor, as functional deficiencies in this pleiotropic kinase predispose both humans and mice to develop various cancer types [5,6]. Indeed, ataxia telangiectasia (AT) patients deficient in ATM have an increased risk of

v2). All sequencing data for this study are publicly available from the GenBank repository (https://www.ncbi.nlm.nih.gov/genbank/) with accession numbers MN251747 (https://www.ncbi.nlm.nih.gov/nuccore/MN251747), MN251748 (https://www.ncbi.nlm.nih.gov/nuccore/MN251748), MN251749 (https://ncbi.nlm.nih.gov/nuccore/MN251749), and MN251750 (https://ncbi.nlm.nih.gov/nuccore/MN251750).

**Funding:** Funded by intramural programs of the National Cancer Institute and the National Institute on Aging, National Institutes of Health.

**Competing interests:** The authors have declared that no competing interests exist

developing multiple cancers [7,8], including T-cell acute lymphoblastic leukemia [9] (T-ALL), a malignancy arising from T-cell precursors arrested at various stages of thymic development [10]. Similarly, ATM-deficient mice (ATMKO) develop progressively enlarging thymi and eventually succumb to thymic T cell lymphoblastic lymphoma (T-LBL), a neoplasm that cytogenetically and developmentally resembles human T-ALL [11,12]. Notably, murine ATMKO thymic T-LBLs possess characteristic chromosomal 14 translocations [13,14] or amplifications [12] involving the TCRα/δ locus, syntenic to the inv(14;14) or 14q11 translocations that are frequently detected in human T-ALL [15,16]. Additionally, both human T-ALLs and murine ATMKO thymic T-LBLs express marked heterogeneity in their developmental profiles, including in gene rearrangement and expression of the T-cell receptor [12,17,18]. These similarities mark the potential of ATMKO thymic T-LBL to serve as an animal model for human T-ALL.

Owing to recent advances in intensive multiagent chemotherapy regimens, ALL patients have attained significantly improved initial complete response rates [19]. However, despite their high efficacies, current regimens can yield delayed side effects including cardiotoxicity [20] and metabolic syndromes [21]. Furthermore, refractory or relapsed disease still poses a significant clinical challenge—five-year survival in relapsed patients for both younger and adult T-ALL patients are strikingly dismal at 23% [22] and 7% [23], respectively. Hence, there remains a need to identify and target pathways specific to these leukemia cells to improve long-term patient survival and potentially mitigate treatment side effects.

One such possible target amenable to therapeutic intervention is the PI3K/AKT/mTOR axis. The PI3K/AKT/mTOR pathway is a ubiquitous major regulator of cellular glucometabolism, apoptosis, and protein synthesis [24] and is frequently dysregulated in human T-ALL [25]. PTEN is a lipid phosphatase that antagonizes the PI3K-dependent phosphorylation of $PtdIns_{4,5}P_2$ ($PIP_2$) into $PtdIns_{3,4,5}P_3$ ($PIP_3$) needed for optimal activation of the AKT/mTOR-mediated pro-survival program [26] and is also one of the most frequently inactivated components of this pathway in T-ALL [27–30]. Mechanisms of PTEN inactivation in T-ALL have been found to be variable and include genomic mutations/deletions and post-translational modification [28,30,31]. The importance of PTEN in negatively regulating PI3K/AKT/mTOR signaling in T-cell precursors is directly demonstrated in mice with T-cell-specific deletion of PTEN (Lck-Cre or CD4-Cre x Pten fl/fl), whose thymocytes have increased AKT phosphorylation at Ser473, are hyperplastic, and are susceptible to malignant transformation into lymphoma [32].

In parallel to the *PTEN* mutations frequently observed in human T-ALL, a previous study of ATMKO thymic T-LBLs reported frequent loss of a locus on chromosome 19 containing the *Pten* gene as detected by aCGH [12]. These *Pten* alterations were not completely characterized, and importantly, it was not clear if these alterations affected PI3K/AKT/mTOR signaling or contributed to ATMKO T-LBL survival, which would be critical to identifying a potential pathway for therapeutic targeting.

Here, we directly investigated the roles of PTEN and PI3K/AKT/mTOR signaling in the context of ATM deficiency. Genetic and protein expression studies of murine ATMKO thymic T-LBL cells revealed that nearly all lymphomas tested possessed genetic *Pten* alterations that resulted in the loss of PTEN expression and/or function. Correlating with the loss of PTEN, ATMKO T-LBLs exhibited constitutively increased AKT (S473) phosphorylation as compared to non-malignant thymocytes. Importantly, all ATMKO thymic T-LBLs were sensitive to treatment with the small molecule AKT inhibitor MK-2206, indicating their strong dependence on AKT signaling for survival. Although there appears to be a selective pressure to lose PTEN in ATMKO T-LBLs, our observation that PTEN is expressed at wild-type levels in non-malignant ATMKO thymocytes demonstrates that ATM deficiency in and of itself does not cause global loss of PTEN expression. Altogether, these results identify AKT signaling as an oncogenic

target in ATMKO thymic T-LBL and provide further rationale to investigate inhibitors of the PI3K/AKT/mTOR pathway, particularly AKT inhibitors, in the treatment of human T-ALL.

## Materials and methods

### Mice breeding

ATMKO mice were bred by intercrossing ATM(+/-).129 (IMSR_JAX:008536) and maintained at Frederick Cancer Research Facility (FCRF) (Frederick, MD) following protocols approved in writing by the NCI and FCRF Institutional Animal Care and Use Committee (IACUC). Necropsy of mice was performed after the detection of significant morbidity (i.e. rapid/shallow breathing) or spontaneous death similarly to as previously described [33], and enlarged thymic masses were harvested to establish primary cell lines.

### Primary cell lines

ATM-deficient T and B cell lymphoma cells were harvested from thymic and splenic masses, respectively, before being cultured in tissue culture (TC) medium consisting of RPMI 1640 (Lonza), 50uM 2-mercaptoethanol (Fisher Scientific), 1mM sodium pyruvate (Corning), 1X MEM NEAA (Corning), 10mM HEPES buffer (Corning), 1X Pen-Strep/L-glutamine (Gibco), and 10% heat-inactivated fetal calf serum (Gemini-Bio). Analysis of the B cell lymphomas utilized in this study was published previously [34].

### Genomic PCR for *Pten* exome

DNA was extracted from tumor and control cells using the Promega Wizard Genomic DNA purification kit using the "Isolation of Genomic DNA from Animal Tissue and Tissue Culture Cells" protocol. 25 ng DNA was used for PCR experiments (94˚C 2 min, 94˚C 30 sec, 55˚C 30 sec, 72˚C 1 min cycled 35x) and subsequent products were resolved on 1.5% agarose gel using 9 *Pten* primers previously described [35]. IgM positive control forward primer: 5'-CCG TCT AGC TTG AGC TAT TAG G-3'. Reverse primer: 5'-GAA GAG GAC GAT GAA GGT GG3'.

### *Pten* full-length cDNA PCR

5–10 million cells were suspended in 600μl of buffer RLT and 2-mercaptoethanol before RNA was extracted (QIAGEN RNeasy Mini Kit). cDNA was synthesized using 1000 ng RNA and oligo(dT) (Invitrogen SuperScript III First-Strand Synthesis System for RT-PCR, 50˚C 50 min, 85˚C 5 min, 37˚C 20 min.) The PTEN mRNA sequence NM_008960.2 was used for construction of the *Pten* cDNA primers. Forward primer: 1F – GCCAAGTCCAGAGCCATTTC (Pos: +770–789). Reverse primer: 5R – ACAAGATTGGTCAGGAGAAGAGA (Pos: +2218–2240). Length of product: 1469 bp. 15 ng cDNA of each sample was used for PCR amplification (94˚C 2 min, 94˚C 30 sec, 58˚C 30 sec, 72˚C 1 min cycled 30X, 72˚C 10 min). B-actin control forward primers: 5'-ATG CCA ACA CAG TGC TGT CTG GTG G-3'. Reverse primers: 5'-CTG ATC CAC ATC TGC TGG AAG GTG-3'.

### Sanger sequencing

The *Pten* coding sequence with overhang was amplified by extracting RNA and reverse transcribing to cDNA as above. cDNA was purified (QiaQuick PCR Purification Kit) and ligated into a linear pCR2.1 vector and circularized into plasmid form (Invitrogen TA Cloning Kit with pCR2.1 Vector and One Shot TOP10 Chemically Competent E. Coli) in 3-to-1 PCR product-to-vector mass ratio. The vectors were cloned into TOP10 E. coli and 10 colonies were selected from which DNA was extracted (QIAprep Spin Miniprep Kit). Purified plasmids were

sequenced by the Sanger method using four independent primers: M13R (5'-`CAGGAAA CAGCTATGAC`-3'), M13F (5'-`GTAAAACGACGGCCAG`-3'), Pten3F (5'-`AGGCACAAGAGG CCCTAGAT`-3'), and Pten4R (5'- `TGCTAGCCTCTGGATTTGATGG`-3'). Sequence data was analyzed by DNASTAR SeqMan Pro 13 and aligned to the reference *Pten* mRNA sequence (NM_008960.2). Sequencing data was submitted to GenBank (MN251747-MN251750).

## PCR to determine micro mutations in tumors 46500 and 45190

DNA from tumors 45190 and 46500 was amplified using primers flanking the putative mutation sites of interest (Pten1 and Pten4, respectively) listed below. 1µg DNA of 45190 was sequentially digested with the PstI enzyme and then BstZI/I-HF enzymes. 1µg DNA of 46500 was digested with the RsaI and BfuCI enzymes. All enzymes from ThermoFisher. Products were run on 20% polyacrylamide gel and visualized.

Pten1F: 5'- `GCCAAGTCCAGAGCCATTTC`-3'
Pten1R: 5'- `ATGTCTCTCAGCACATAGATTGT`-3'
Pten4F: 5'- `GTACTTTGAGTTCCCTCAGCCA`-3'
Pten4R: 5'- `TGCTAGCCTCTGGATTTGATGG`-3'

## Flow cytometry

For PTEN staining, lymphoma cells were first permeabilized and fixed using the BD Biosciences Cytofix/Perm Buffer system and following the manufacturer's instructions before being stained with either PTEN-specific (PE mouse anti-PTEN A2B1, BD Biosciences) or control (PE mouse IgG1 kappa isotype control 551436, BD Biosciences) antibodies for 30 minutes. For staining thymocytes, cells were fixed and permeabilized, stained with FITC anti-CD4 and APC anti-CD8 (BD Biosciences) for 30 minutes, washed 3x in BD Perm/Wash buffer, and then incubated with PTEN- or control antibodies. After staining, all cells were washed 3X in Perm/ Wash buffer and filtered. Data were collected with FACS Calibur (BD Biosciences) and analyzed with FlowJo software (Tree Star). For comparing PTEN expression levels between ATMKO and ATMWT non-malignant thymocytes, ΔMFI was calculated by: MFI(PTEN antibody)–MFI(Control antibody).

For pAKT and AKT staining, lymphoma and thymocyte cells were permeabilized as described above and resuspended in BD Phosflow Perm/Wash Buffer 1. Cells were then stained with either AKT-specific (PE rabbit anti-AKT C67E7, Cell Signaling), pAKT (S473)-specific (PE rabbit anti-pAKT (S473) D9E, Cell Signaling), or control antibody (PE rabbit IgG isotype control DA1E, Cell Signaling) for 30 minutes. After staining, all cells were washed 3X in perm/wash buffer and filtered. Data were collected with FACS Calibur (BD Biosciences) and analyzed with FlowJo software (Tree Star). For comparing AKT and pAKT expression levels between non-malignant thymocytes and ATMKO T-LBLs, ΔMFI was calculated by: MFI (pAKT(S473) or AKT antibody)–MFI(Control antibody).

## Western blot

Lymphoma cells were lysed in buffer containing 50 μM Tris (pH 7.4), 150 μM NaCl, 1 mM $Na_2VO_4$, 1% Nonidet P-40, and protease inhibitor mixture. Protein lysates were used for biochemical analysis as was published previously [36]. Antibodies used include Rabbit PTEN mAB #9188 (Cell Signaling) specificity for C-terminal human PTEN (1:1000) and Rabbit HRP (1:2000, Southern Bio).

### *In vitro* cell inhibition experiments

$5x10^3$ cells were plated in quadruplicate replicates in 96-well flat-bottom plates (Costar) in 100μl of TC medium (RPMI 1640, 10% FBS, Pen/Strep, Glutamine, 2-mercaptoethanol) and incubated with TC medium-diluted inhibitory reagent, MK-2206 (solubilized in DMSO) or TC-medium-diluted DMSO for 48 hours. After 48 hours, 20μl of CellTiter (96 Aqueous One Solution Cell Proliferation Assay Promega) was added to each well, and plates were incubated for 2 additional hours at 37˚C. To determine cell recovery plates were read by colorimetric analysis at 490nm wavelength. Percent inhibition equation:

$$\left( 1 - \frac{Absorption\ of\ cells\ with\ MK2206\ concentration\ X - Absorption\ of\ DMSO\ only}{Absorption\ of\ cells\ with\ DMSO\ concentration\ X - Absorption\ of\ DMSO\ only} \right) x\ 100$$

### Statistical analysis

For comparison of expression levels of AKT, pAKT, pAKT/AKT, and PTEN as detected by flow cytometry, Student's t test with two-tailed distributions was performed for statistical analyses using Microsoft Excel. For comparison of percent inhibition of cell viability with MK-2206, statistical significance was assessed by a mixed linear regression analysis with log transformed MK-2206 concentrations and cell types as covariates using R Statistical Software (v4.1.1; R Core Team 2021). $P<0.05$ was defined as the threshold to meet statistical significance.

## Results

### ATMKO thymic T-LBLs exhibit genetic alterations in *Pten*

In this study, we examined murine ATMKO thymic T-LBLs after adaptation into *in vitro* culture as a model to identify oncogenic pathways that may be relevant therapeutic targets for human T-ALL. It was previously reported that a subset of murine ATMKO T-LBLs exhibit homozygous deletions in *Pten* (6/18 cases) [12]. These mutations were not characterized, and it was not determined whether these alterations affected PI3K/AKT/mTOR signaling or contributed to ATMKO T-LBL survival. We therefore directly assessed the status of the *Pten* gene and the PI3K/AKT/mTOR axis in ATMKO thymic T-LBLs generated in our colony.

We first used PCR-based strategies to compare the *Pten* exomes of these T-LBL cells to both normal mouse splenocytes and to ATMKO B cell lymphomas found to be dependent on BCR/ NF-*k*B signaling [34]. Genomic DNA was amplified using primer sets that flank each of the 9 *Pten* exons from 12 ATMKO T-LBLs (Fig 1A), control splenocytes and control ATMKO B cell lymphoma (178119). No PCR products corresponding to any of the 9 *Pten* exons were detected in 3/12 T cell tumors (43224, 334967, and 49733), suggesting homozygous deletion of the entire *Pten* exome in these lymphomas. Additionally, 2/12 tumors (53466 and 45220) showed PCR amplification of exons 2–9 but not exon 1, while 1/12 tumors (43225) was PCR-positive for exons 1–6 but not exons 7–9. For the remaining 6/12 ATMKO T-LBLs (46500, 45190, 48440, 51225, 56211, and 57228), PCR analysis amplified products from all 9 *Pten* exons that were indistinguishable in size from the corresponding *Pten* exons amplified from control WT B6 splenocytes and ATMKO B cell lymphoma cells. Together, these results demonstrate that 6/12 ATMKO T-LBLs possess homozygous complete or partial losses of the *Pten* exome while the remaining 6 cell lines possess *Pten* exomes of approximately wild-type length.

We next determined whether the six tumors exhibiting exon-specific genomic PCR products that appear equivalent in size to WT *Pten* exomes (46500, 45190, 48440, 51225, 56211,

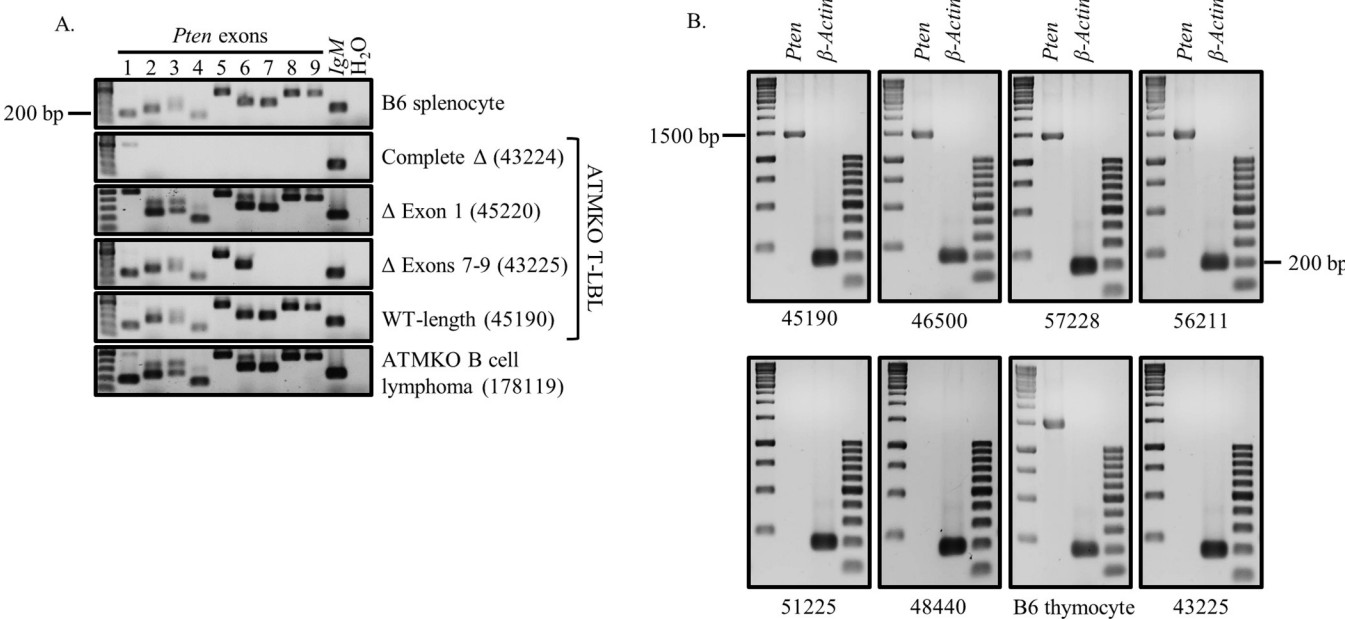

**Fig 1. Multiple ATMKO T-LBLs harbor alterations in Pten exome and fail to transcribe *Pten* mRNA.** (A) Genomic DNA was PCR amplified from B6 splenocytes, ATMKO T-LBLs and an ATMKO B cell lymphoma (178119) (top to bottom) using primers specific for each of the 9 *Pten* exons or control germline IgM. PCR products were resolved on a 1.5% agarose gel and visualized. Results are shown for representative tumors that express the indicated *Pten* genotypes. Complete Δ designates complete deletion of all exons (43224, representative of 334967 and 49733); Δ Exon 1 designates selective loss of exon 1 (45220, representative of 53466); Δ Exons 7–9 designates selective loss of exon 7–9 (43225); and WT-length designates presence of bands equivalent to WT for all exons (45190, representative of 46500, 48440, 51225, 56211, and 57228). (B) cDNA from the 6 ATMKO thymic T-LBLs with WT-length exomes, B6 whole thymus, and tumor 43225 identified to possess an exon 7–9 deletion was PCR-amplified using primers that flanked either the full-length coding sequence of *Pten* (1469 bp) or β-actin (200bp). PCR products were resolved on 1.5% agarose gel and visualized.

and 57228) might nevertheless fail to express a functional RNA transcript. To determine if these six T-LBLs transcribe WT-length *Pten*, we probed cDNA prepared from these lymphomas using primers that flank the entire *Pten* coding sequence.

Control B6 thymus cDNA but not tumor 43225 with a known exons 7–9 deletion amplified full-length *Pten* mRNA PCR product (Fig 1B). Despite the presence of apparent WT-length exomes by genomic PCR in 46500, 45190, 48440, 51225, 56211, and 57228 (Fig 1A), two of these tumors (51225 and 48440) failed to amplify any full-length *Pten* cDNA (Fig 1B). The remaining four tumors (45190, 46500, 56211, and 57228) with apparent WT-length *Pten* exomes also appeared to transcribe WT-length *Pten* cDNA.

Since these gel-based analyses cannot detect small indels and/or balanced mutations, we used Sanger sequencing to analyze cDNA from the entire *Pten* coding region of these four tumors (45190, 46500, 56211, and 57228) with apparent full-length *Pten* cDNA. Sequencing revealed that tumor 46500 contained a single adenine insertion in exon 7, a mutation hotspot for *PTEN* in human T-ALL [27,28], that results in a frameshift mutation and introduces a premature stop codon in exon 8 (NM_008960.2.c.968_969insA). Tumor 45190 was found to contain tandem duplications of a 32-bp sequence in exon 2 that results in a frameshift mutation and introduces premature translational stop codons in the same exon (Fig 2). Tumor 45190 yielded two unique mutated sequences containing the 32-bp duplication in the absence of a wild-type sequence, suggesting the presence of biallelic heterozygous mutations (NM_008960.2.c.125_126ins [125–157] and NM_008960.2.c.158_159ins [127–158]). To corroborate the presence of these frameshift mutations, full-length *Pten* cDNA from tumors 45190 (S1A Fig) and 46500 (S1B Fig) was subjected to PCR-amplification using primers

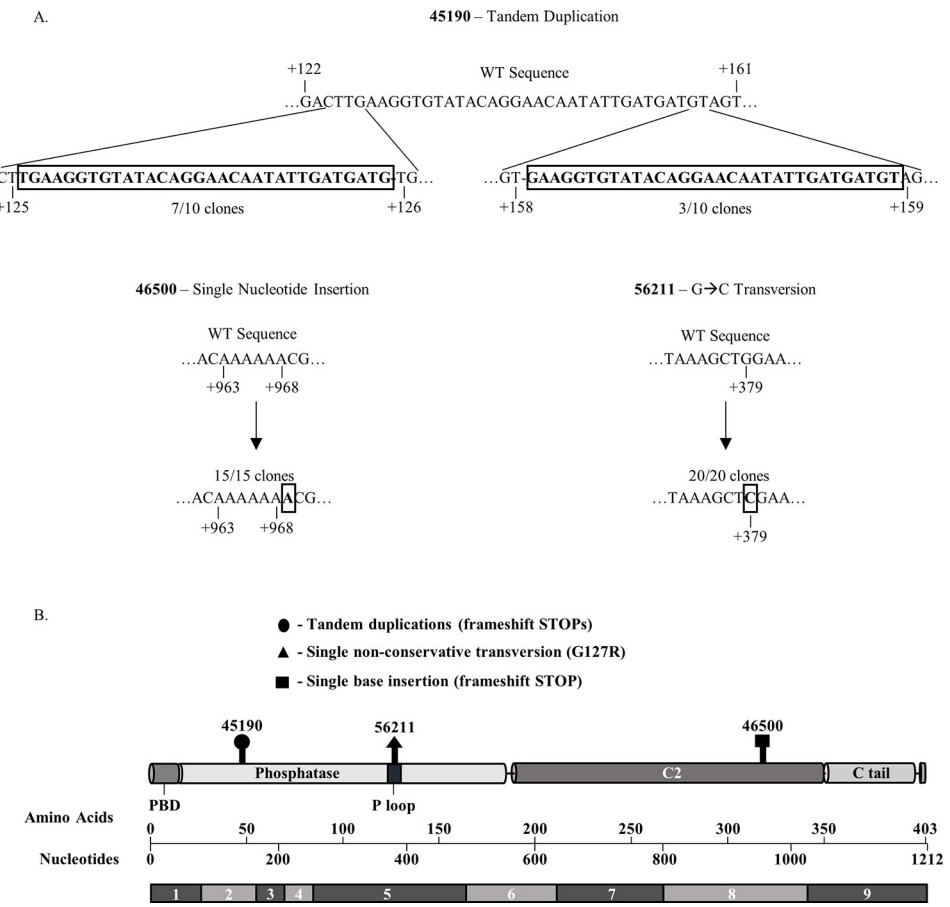

**Fig 2. ATMKO T-LBLs expressing WT-length *Pten* exomes contain *Pten* micro mutations.** (A) Nucleotide wild-type sequences compared to those of tumors 45190, 46500, and 56211 obtained by Sanger sequencing. Sequences with borderline represent inserted nucleotide sequences in tumor 45190, inserted single nucleotide in 46500, and a nucleotide transversion in 56211. (B) Schematic of the murine PTEN protein and corresponding exome. Designated boldface marks identify the locations of *Pten* mutations detected by Sanger sequencing of 3 ATMKO thymic T-LBL samples (Tumor 45190: NM_008960.2.c.125_126ins[125–157] and NM_008960.2.c.158_159ins[127–158]. Tumor 56211: NM_008960.2.c.379G>C. Tumor 46500: NM_008960.2.c.968_969insA).

designed to amplify the putative mutation regions and resolved on polyacrylamide gels to distinguish single base pair differences. Tumor 56211 contained a single G>C point mutation resulting in a G127R amino acid substitution in the catalytically active phosphate-binding loop (P loop) region of the PTEN phosphatase domain [37] (NM_008960.2.c.379G>C). This substitution mutation does not introduce a translational stop codon, but mutation software analyses using SIFT and PROVEAN both predicted the PTEN G127R substitution to be functionally deleterious, scoring 0.01 and -7.42, respectively. Lastly, no mutations in the *Pten* coding region were detected in tumor 57228. Altogether, these results identify a variety of *Pten* deletions and mutations detected in 11/12 (92%) of ATMKO thymic T-LBLs tested that are predicted to impair expression of functional PTEN protein.

## The majority of ATMKO thymic T-LBLs lack PTEN protein expression

We next asked if the *Pten* genomic alterations detected in these lymphomas corresponded to impaired expression of PTEN protein. Flow cytometric analysis of ATMKO T-LBLs using a PTEN C-terminus-specific antibody revealed that 10/12 tumors lacked detectable PTEN

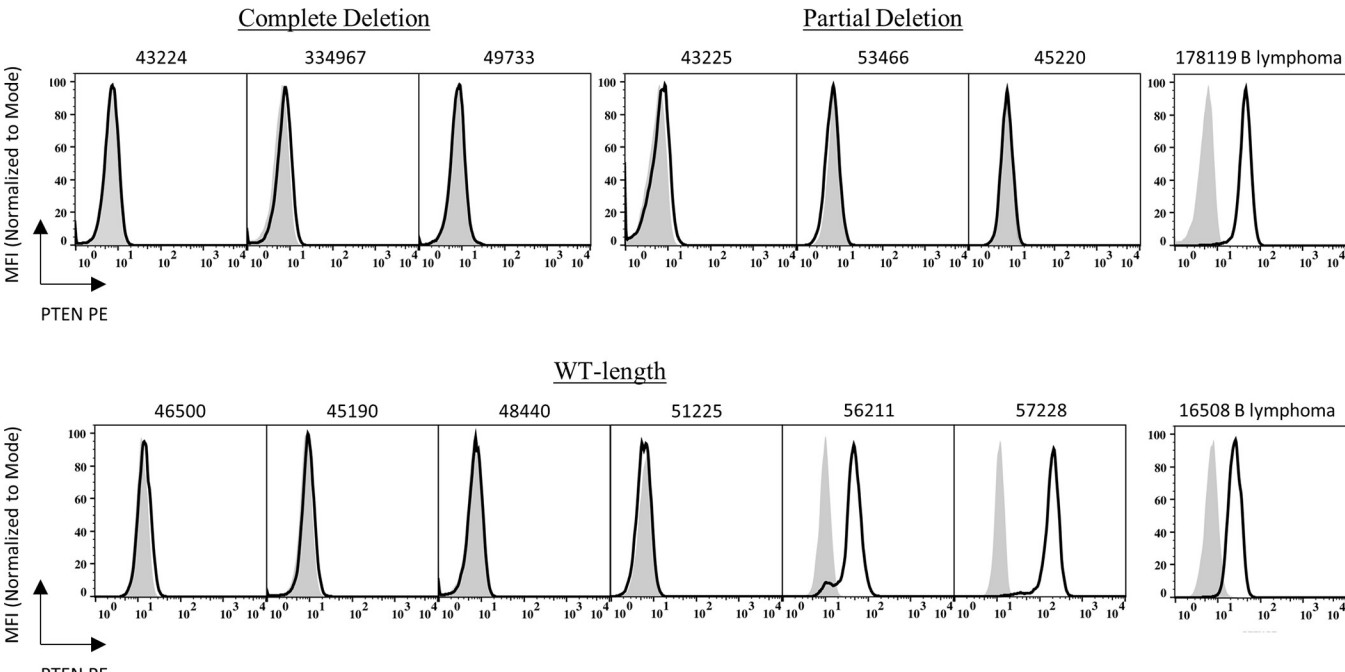

**Fig 3. The majority of ATMKO T-LBLs fail to express PTEN protein.** Flow cytometric analysis of ATM-deficient T cell lymphoma cells with PE-conjugated anti-PTEN (black-lined open histogram) or isotype control (gray shaded histogram) antibody grouped by their respective *Pten* exome profiles. Top row left to right: Staining profiles of ATMKO T-LBL tumors with a complete deletion of the *Pten* exome (43224, 334967, and 49733), a partial deletion of the exome (43225, 53466, and 45220) and an ATMKO B cell lymphoma (178119). Bottom row left to right: ATMKO T-LBL tumors with a WT-length exome (46500, 45190, 48440, 51225, 56211, and 57228) and an ATMKO B cell lymphoma (16508). Each histogram is representative of three to eight experiments.

protein expression (Fig 3). Both tumors 56211, containing a missense mutation in the *Pten* phosphatase region, and 57228, without any identified *Pten* coding mutations, expressed PTEN protein by flow cytometry. In addition, Western blot analysis using an N-terminus specific anti-PTEN antibody also failed to detect PTEN expression in 8/8 tumors analyzed (S2 Fig).

Taken together, these results demonstrate that the majority of ATMKO T-LBLs (10/12) lack detectable PTEN protein expression, supporting the hypothesis that the various *Pten* mutations detected in our samples correspond to the loss of PTEN protein expression.

## ATMKO T-LBLs express constitutively elevated pAKT and are sensitive to treatment with the allosteric AKT inhibitor MK-2206

PTEN functions as a critical negative regulator of the PI3K/AKT/mTOR pathway [24]. Since the majority of our ATMKO T-LBLs exhibit *Pten* genomic alterations and lack detectable PTEN protein, this raised the possibility of the corresponding activation of the PI3K/AKT/mTOR pathway. Therefore, we investigated the status of AKT phosphorylation as a measure of PI3K/AKT/mTOR activation in these tumors. Flow cytometric analyses revealed positive expression of pAKT (S473) in all ATMKO T-LBLs (Figs 4A and S3), including the two tumors 57228 and 56211 that express PTEN protein. Furthermore, pAKT (p<0.001), AKT (p<0.001), and pAKT/AKT (p<0.001) expression were significantly increased in ATMKO T-LBLs as compared to in non-malignant thymocytes (Fig 4B). These results revealed that functional PTEN deficiency in T-LBLs correlated with abnormally increased AKT phosphorylation, a characteristic that could be potentially targeted.

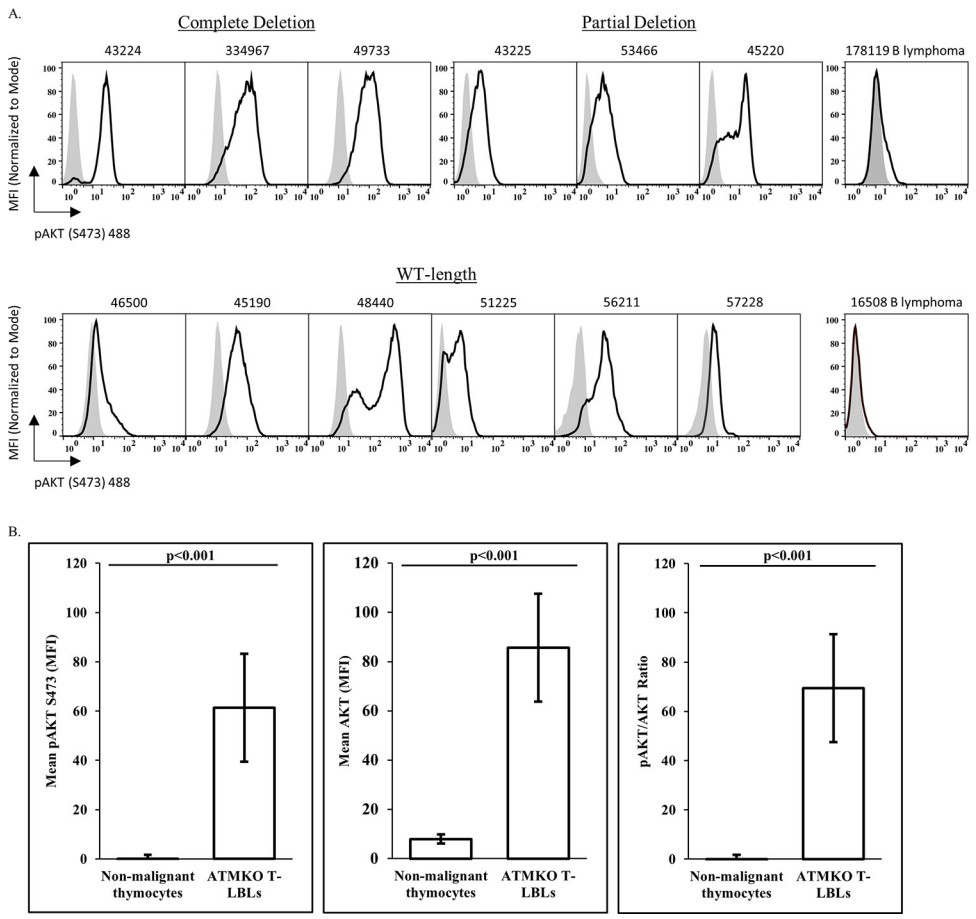

**Fig 4. ATMKO T-LBLs express elevated pAKT.** (A) Flow cytometric analysis of ATM-deficient T cell lymphoma cells with AF488-conjugated anti-pAKT (S473) (black-lined open histogram) or isotype control (gray shaded histogram) antibody grouped by their respective *Pten* exome profiles. Top row left to right: Staining profiles of ATMKO T-LBL tumors with a complete deletion of the *Pten* exome (43224, 334967, and 49733), a partial deletion in the exome (43225, 53466, and 45220) and an ATMKO B cell lymphoma (178119). Bottom row left to right: ATMKO T-LBL tumors with a WT-length exome (46500, 45190, 48440, 51225, 56211, and 57228) and an ATMKO B cell lymphoma (16508). Each histogram is representative of three to twelve experiments. (B) Mean expression of pAKT (left), total AKT (middle) and pAKT/AKT ratio (right) in non-malignant thymocytes (n = 7 mice, 3 ATMKO, 1 ATMHET, 3 ATMWT) and ATMKO T-LBLs (n = 9 tumors). Data shown are the mean +/- SEM. Statistical significance was assessed by two sample two-tailed Student's t-test (p values <0.05 were considered statistically significant).

The demonstrated absence of PTEN protein in the majority of these T-LBLs and the concomitant activation of AKT in all these T cell lymphomas raised the possibility that these T cell tumors require PI3K/AKT/mTOR signaling for their survival. MK-2206 is a well-described selective small molecule pharmacological inhibitor of AKT [38] that reduced pAKT (S473) expression in ATMKO T-LBL (S4 Fig, p = 0.0028). ATMKO T-LBLs and control ATMKO B cell lymphomas were cultured *in vitro* with MK-2206 or vehicle control DMSO for 48 hours and then assayed for viable cell recovery. Treatment with MK-2206 resulted in decreased viable cell recovery for all ATMKO thymic T-LBLs without significantly inhibiting viability of NF-*k*B-dependent ATMKO B cell lymphomas [34] (Fig 5A and S1 Appendix, p<0.001). Importantly, tumors 56211 (which was found to have a mutation in the PTEN phosphatase loop predicted to be deleterious) and 57228 (in which no *Pten* coding mutation was detected) expressed pAKT and were sensitive to MK-2206 despite expressing PTEN (Fig 5B). In all, we

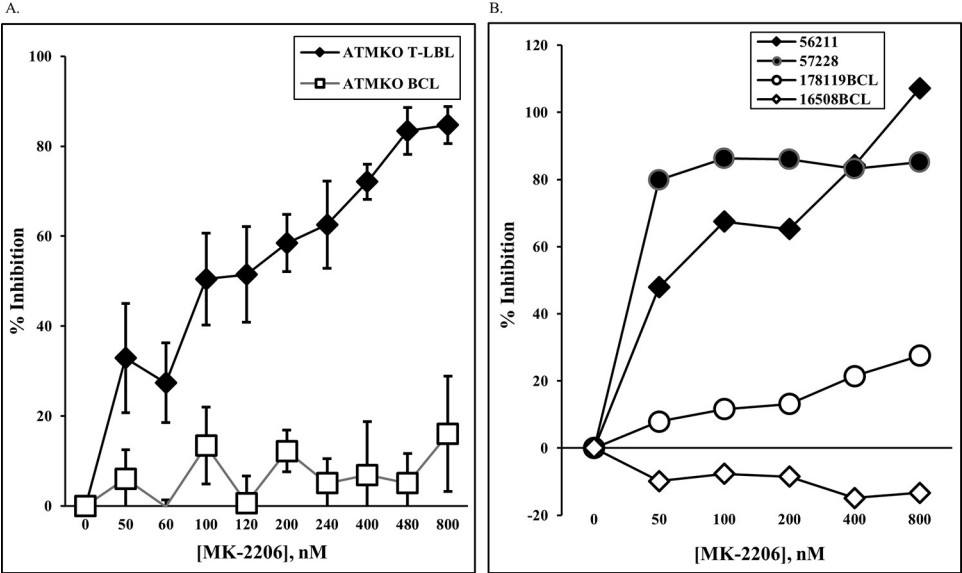

**Fig 5. Viable cell survival of ATMKO T-LBLs is impaired by treatment with MK-2206, an AKT inhibitor.** (A) ATMKO T (closed diamond) and B (open square) cell lymphomas were incubated with titrated concentrations MK2206 or diluent DMSO for 48-hrs and subsequently tested for cell viability using Cell-titer, a colorimetry-based assay. Percent inhibition of viable cell recovery was calculated using DMSO-only controls. Data shown are the mean +/- SEM of 5 independent experiments containing 7 ATMKO T and 2 B cell tumors across multiple concentrations of MK-2206. Statistical analysis was performed by a linear mixed model (p values <0.05 were considered to represent a statistically significant interaction effect between cell types and log transformed MK-2206 concentrations). (B) Representative experiment indicating % inhibition of cell viability in individual PTEN-positive ATMKO T-LBLs (56211 and 57228, filled) and ATMKO B cell lymphomas (178119 and 16508, open). BCL denotes B cell lymphoma.

demonstrate that all ATMKO T-LBLs express increased AKT phosphorylation and are sensitive to a pharmacological AKT inhibitor, highlighting their dependence on AKT signaling for survival.

## Non-malignant ATMKO thymocytes express ATMWT levels of PTEN throughout thymic development

The high prevalence of PTEN deficiency in our tumors led us to question whether loss of PTEN in thymocytes occurs as a consequence of ATM deficiency itself, independent of neoplastic transformation. Flow cytometry using anti-PTEN antibodies showed that although the majority of ATMKO thymic T-LBLs failed to express PTEN, expression levels of PTEN between non-malignant ATMKO thymocytes and ATMWT thymocytes were equivalent (Fig 6A, p = 0.418). Furthermore, we observed no significant difference in PTEN expression between ATMKO and ATMWT thymocytes across all CD4/CD8 developmental subsets (DN (CD4⁻ CD8⁻, p = 0.116), DP (CD4⁺ CD8⁺, p = 0.306), CD4SP (CD4⁺ CD8⁻, p = 0.190), or CD8SP (CD4⁺ CD8⁻, p = 0.152), Fig 6B). These results demonstrate that ATM expression is not required for PTEN expression in ATMKO thymocytes, suggesting that PTEN loss is instead an oncogenic event selected for during ATMKO thymic lymphomagenesis.

## Discussion

In this study, we characterize the mechanisms of recurrent PTEN inactivation in the ATM-deficient mouse model of human T-ALL and importantly describe the concomitant constitutive activation of AKT (Table 1 for summary). The majority of these T-LBLs exhibit a variety

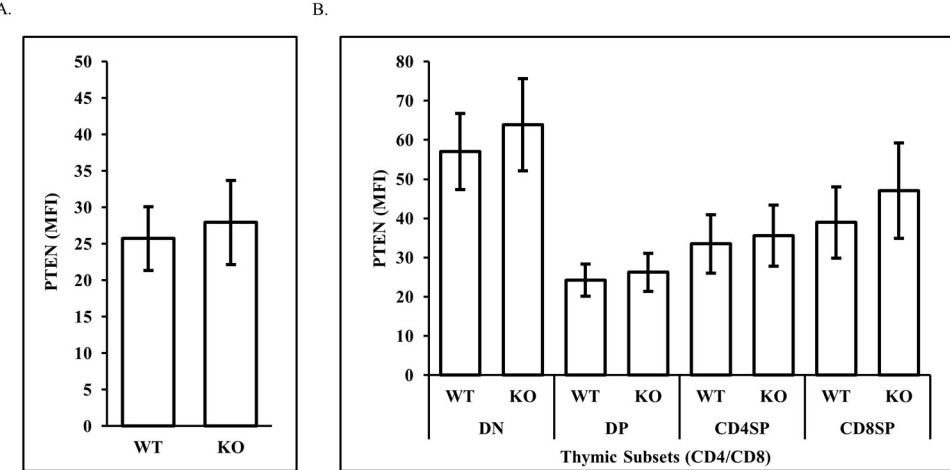

**Fig 6. ATMKO non-malignant thymocytes express levels of PTEN protein equivalent to their ATMWT counterparts.** As assessed by flow cytometry, PTEN protein expression is equivalent between total ATMWT and -KO non-malignant thymocytes (A) and across thymic developmental subtypes (B). Data are the mean ΔMFI (MFI PTEN– MFI control antibodies) ± SEM of PTEN expression acquired by flow cytometric analysis of 3 independent experiments. Statistical significance was assessed by paired two sample two-tailed Student's t-test analysis (p values <0.05 were considered statistically significant).

**Table 1. Summary analysis of ATMKO T-LBLs.**

| Mouse ID# | Full-length *Pten* cDNA (PCR) | Sanger Sequencing PTEN cDNA (Poly A+) | PTEN protein (flow cytometry; anti-C terminus) | PTEN protein (Western blot; anti-N terminus) | pAKT (S473) + or– (flow cytometry) | Sensitive to pAKT inhibition (MK-2206) |
|---|---|---|---|---|---|---|
| **Complete Deletion of PTEN exome (genomic PCR)** | | | | | | |
| 43224 | — | ND | — | — | + | Yes |
| 334967 | — | ND | — | — | + | Yes |
| 49733 | — | ND | — | — | + | Yes |
| **Partial Deletion of PTEN exome (genomic PCR)** | | | | | | |
| 43225 (Δ exons 7–9) | — | ND | — | — | + | Yes |
| 53466 (Δ exon 1) | — | ND | — | ND | + | ND |
| 45220 (Δ exon 1) | — | ND | — | ND | + | ND |
| **WT-size PTEN exome (genomic PCR)** | | | | | | |
| 46500 | + | Single base insertion (frameshift STOP) | — | — | + | Yes |
| 45190 | + | Tandem mutation (frameshift STOP) | — | — | + | Yes |
| 48440 | — | ND | — | — | + | Yes |
| 51225 | — | ND | — | ND | + | ND |
| 56211 | + | G→C transversion in P loop of phosphatase domain | + | ND | + | Yes |
| 57228 | + | No coding mutations detected | + | ND | + | Yes |

ND = not done. Table summarizes experimental findings for the 12 ATMKO T-LBLs studied.

of *Pten* gene alterations that result in the loss of functional protein. All samples express pAKT and are sensitive to treatment with the pan-AKT inhibitor MK-2206, demonstrating a tumor vulnerability amenable to pharmacological targeting.

We identified detectable genomic *Pten* alterations in 11/12 mouse ATMKO T-LBLs. It is striking that these alterations are widely distributed throughout the *Pten* genome and included partial or whole exome deletions, single nucleotide insertions/substitutions, and duplications without any apparent predisposition for a particular site. This finding suggests that ATMKO T-LBLs experience strong selective pressure to inactivate PTEN. In human T-ALL, *PTEN* alterations preferentially cluster in exon 7 [28] but also commonly occur throughout the gene [39]. Although *PTEN* alterations in human T-ALL have been described [27,40,41], only a few clinical trials have investigated the efficacy of targeting the PI3K/AKT/mTOR pathway. These trials include other hematological malignancies, a characteristic that makes interpretation of these results for T-ALL difficult to make. The lack of more extensive T-ALL-specific clinical trials has been due, at least in part, to the historically unclear clinical impact of *PTEN* alterations on the prognosis of T-ALL patients. However, one recent study of 573 pediatric and adult T-ALL patients [42] reported that large deletions, and not small deletions/mutations, in *PTEN* significantly predicted poor 5-year overall survival (OS) and disease-free survival (DFS) rates.

It has been reported that genomic deletions of a single gene can result in reproducible spontaneous mutations in one or more additional genes [43]. We therefore considered the possibility that *Pten* mutations, which occurred at high frequency (92%) in our T-LBLs, might be the direct result of ATM deficiency in our mice. However, we observed that ATM deficiency per se did not lead to global loss of PTEN expression, as non-malignant ATM-deficient thymocytes express wild-type levels of PTEN protein and have undetectable pAKT levels. It is yet to be determined in ATMKO T-LBLs if functional inactivation of PTEN is required for lymphomagenesis or is a mutational hit selected for post-transformation, but studies of primary human T-ALL samples using single-cell DNA amplicon sequencing of diagnosis-remission paired samples and xenografts have suggested that PTEN loss may be a relatively late event associated with subclonal evolution and relapsed disease [44–46].

In our MK-2206 inhibition study, all T-LBLs tested were sensitive to treatment, including tumors 56211 and 57228 that expressed PTEN protein. While the sensitivity of tumor 56211 to MK-2206 can be explained by the mutation detected in the PTEN phosphatase loop, the inhibition of tumor 57228 (no *Pten* coding mutations detected) raises the possibility of an indirect functional inactivation of PTEN. A study of sporadic primary human T-ALL samples reported that of those expressing PTEN, the phosphatase activity of most samples was being suppressed at the post-translational level by the CK2 kinase [31]. Although the predominant mechanism of PTEN inactivation may differ between human T-ALL and murine ATMKO T-LBL, 88% of these primary human samples demonstrated evidence of AKT activation, suggesting that there is a similar convergence towards AKT activation between human and murine T-ALL/LBL. Therefore, assessing PTEN status without also directly measuring AKT activity in human T-ALL may underestimate the potential clinical benefit of targeting AKT signaling.

To date, clinical trials targeting the PI3K/AKT/mTOR pathway specifically in T-ALL are few. One encouraging phase I/II trial combining the mTORC1 inhibitor everolimus with the hyperCVAD chemotherapy regimen found that 5/10 heavily pre-treated T-ALL patients demonstrated clinically-meaningful responses [47], but additional trials dedicated to testing the efficacy of targeting PI3K/AKT/mTOR specifically in T-ALL are needed. While AKT inhibitors are being evaluated for efficacy in multiple cancers [48], they are yet to be tested for their effectiveness in T-ALL patients. The normal expression of PTEN and the lack of detectable AKT phosphorylation in non-malignant whole thymocytes suggest that use of AKT inhibitors in T-ALL therapy could largely preserve the healthy hematopoietic population. A pre-clinical

study supports the potential for an acceptable therapeutic index with this inhibitor, as MK-2206 treatment was cytotoxic to primary pediatric T-ALL cells *ex vivo* without significantly affecting the viability of normal CD4[+] peripheral T cells and CD34[+] hematopoietic progenitor cells [49]. AKT inhibition has also been demonstrated to increase sensitivity of T-ALL samples to standard corticosteroid therapy [50,51] as well as to the salvage agent nelarabine [52].

Overall, the study presented here identifies PTEN inactivation as an oncogenic event in immature murine thymic T cell lymphomas with ATM deficiency and highlights AKT as a potential molecular target for therapy in human T-ALL/T-LBL. This mouse model may be useful in studying therapies for human immature T cell malignancies, particularly in the context of PI3K/AKT/mTOR dysregulation and/or ATM deficiency.

## Supporting information

**S1 Fig. Confirmation of *Pten* micro mutations in tumors 45190 and 46500.** PCR-amplified and restriction enzyme-digested cDNA of tumors 45190 (A) sequenced to have a 32-bp duplication (PstI and BstZI/I-HF enzymes) and 46500 (B) sequenced to have a single-base insertion (RsaI and BfuCI enzymes) were resolved on a 20% polyacrylamide gel. Numbers 1–10 represent technical replicates of each sample and B6 represents wildtype B6 thymus DNA.
(TIF)

**S2 Fig. PTEN deficiency of ATMKO T-LBLs by Western blot.** Protein lysates of B6 splenocytes and ATM-deficient T/B cell lymphomas were immunoblotted using antibodies specific for C-terminus of PTEN (Top row) or for β-Actin (bottom row).
(TIF)

**S3 Fig. AKT expression in T-LBL.** Flow cytometric analysis of ATM-deficient T and B cell lymphomas with PE-conjugated anti-total AKT (black-lined open histogram) or isotype control (gray shaded histogram) antibody grouped by their respective *Pten* exome profiles. Top row left to right: Staining profiles of ATMKO T-LBL tumors with a complete deletion of the *Pten* exome (43224, 334967, and 49733), a partial deletion of the exome (43225, 53466, and 45220) and an ATMKO B cell lymphoma (178119). Bottom row left to right: ATMKO T-LBL tumors with a WT-length exome (46500, 45190, 48440, 51225, 56211, and 57228) and an ATMKO B cell lymphoma (16508). Each histogram is representative of three to twelve experiments.
(TIF)

**S4 Fig. MK2206 inhibits pAKT (S473) expression in T-LBL.** Tumor 43224 mean expression of pAKT (S473) was compared between DMSO control and 200nM MK-2206 after 30 minutes of incubation with +/- SEM across 3 independent experiments. Statistical analysis was performed by using a two sample two-tailed Student's t-test assuming unequal variances (p values <0.05 were considered statistically significant).
(TIF)

**S1 Appendix. Full regression analysis report for Fig 5.**
(DOCX)

## Acknowledgments

We thank Drs. Nan Ping Weng and Thomas Nguyen for guidance on sequence submission. We thank Dr. Ying Lu for aiding in flow cytometry experiments. We thank Dr. Arthur Shaffer for critical reading of the manuscript and insightful comments.

## Author Contributions

**Conceptualization:** Joseph B. An, Karen S. Hathcock, Richard J. Hodes.

**Data curation:** Joseph B. An, Karen S. Hathcock, Seth M. Steinberg, Hyoyoung M. Choo-Wosoba.

**Formal analysis:** Joseph B. An, Karen S. Hathcock, Seth M. Steinberg, Hyoyoung M. Choo-Wosoba, Richard J. Hodes.

**Investigation:** Joseph B. An, Karen S. Hathcock.

**Methodology:** Joseph B. An, Karen S. Hathcock.

**Supervision:** Richard J. Hodes.

**Writing – original draft:** Joseph B. An, Karen S. Hathcock.

**Writing – review & editing:** Joseph B. An, Karen S. Hathcock, Richard J. Hodes.

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
