## [Decision Letter · Decision Letter 0]

21 Jun 2024

PONE-D-24-21090ATM-deficient murine thymic T-cell lymphoblastic lymphomas are PTEN-deficient and require AKT signaling for survivalPLOS ONE

Dear Dr. Hodes,

Thank you for submitting your manuscript to PLOS ONE. After careful consideration, we feel that it has merit but does not fully meet PLOS ONE’s publication criteria as it currently stands. Therefore, we invite you to submit a revised version of the manuscript that addresses the points raised during the review process.

We look forward to receiving your revised manuscript.

Kind regards,

Alvaro Galli

Academic Editor

PLOS ONE

Journal Requirements:

"Funded by intramural programs of the National Cancer Institute and the National Institute on Aging, National Institutes of Health"

3. Please note that funding information should not appear in the Acknowledgments section or other areas of your manuscript. We will only publish funding information present in the Funding Statement section of the online submission form. Please remove any funding-related text from the manuscript.

Reviewers' comments:

Reviewer's Responses to Questions

**Comments to the Author**

1. Is the manuscript technically sound, and do the data support the conclusions?

Reviewer #1: Yes

Reviewer #2: Yes

Reviewer #3: Yes

2. Has the statistical analysis been performed appropriately and rigorously? 

Reviewer #1: Yes

Reviewer #2: Yes

Reviewer #3: Yes

3. Have the authors made all data underlying the findings in their manuscript fully available?

Reviewer #1: Yes

Reviewer #2: Yes

Reviewer #3: Yes

4. Is the manuscript presented in an intelligible fashion and written in standard English?

Reviewer #1: Yes

Reviewer #2: Yes

Reviewer #3: Yes

5. Review Comments to the Author

Reviewer #1: The study titled ATM-deficient murine thymic T-cell lymphoblastic lymphomas are PTEN-deficient and require AKT signaling for survival by An et al delves into the genetic and signaling aspects of T-cell lymphoblastic lymphoma cells from ATM-deficient mice, a model mimicking human T-ALL. The authors show that most ATM-deficient thymic T-LBLs exhibit genetic alterations in Pten, leading to the loss of PTEN expression and increased AKT activation. Furthermore, they show that all ATM-deficient T-LBLs are sensitive to AKT inhibition, suggesting the potential therapeutic potential of AKT inhibitors in treating human T-ALL.

Although this study is conceptually interesting, the presented results are rather weak.

Furthermore, several questions arise while reading the manuscript and the authors may address them during the revision process:

- The first sentence of the abstract starts with the words “mice deficient in ATM…”. This is misleading as one immediately assumes that the experiments in this study are conducted in vivo. This is, however, not the case as the data in the manuscript are exclusively derived from cell culture-based experiments. In the Methods section, it is furthermore unclear, how the cells were extracted from mice to generate the analyzed cell lines.

- The figures are pixelated and difficult to decipher

- The figure captions are somehow embedded in the text and are easily missed.

- In Figure 6A, define non-malignant versus malignant cells

- In reference to Figure 6, would deleting Pten, e.g. using CRISPR-Cas9, in the PTEN-sufficient T-LBL cells mimic the phenotype observed in the PTEN-deficient cell lines?

- What are the functional properties of the malignant T-LBL cells? Are they more proliferative?

- Would these cells be resistant to standard treatment of, e.g., T-ALL?

- Are the T-LBL cells responsive to Everolimus treatment?

Reviewer #2: The authors reported the frequent pTEN deficiency either by complete deletion, partial deletion or mutations of PTEN, inducing pAKT on Ser 473 and sensitivity to inhibitors of AKT signaling among the 12 ATMKO mice with T-LBL in this series.

Minor comments

1- The authors should explain how they select these 12 ATMKO T-LBL compared to their littermate of ATMKO mice and ATMKO with T LBL non selected for this study

2-the p values for figure 6B should be given

Reviewer #3: In this manuscript by An et al., the authors confirm a previous discovery that ATMKO T-LBLs tend to exhibit Pten genetic alterations. The study characterized the mutations and found that most ATMKO T-LBLs have various genomic alterations in the Pten gene, resulting in the loss of functional PTEN protein. Consistently, all T-LBLs in the study demonstrated higher levels of phosphorylated AKT (pAKT), indicating the activation of AKT signaling. Importantly, this pAKT signaling appears to be crucial for the survival of the lymphomas, as indicated by the data on the AKT inhibitor MK-2206. The experiments were well-designed, and the results seem robust. This work thus nicely documents an interesting finding.

I have no criticism except that in Figures 1, 3, 4, and 5, the authors only present data from representative ATMKO T-LBLs. Considering there are only 12 of them and not much mechanistic insight was demonstrated in this work, I think that all the data should be included in this manuscript, at least in the supplementary materials.

6. PLOS authors have the option to publish the peer review history of their article (what does this mean?). If published, this will include your full peer review and any attached files.

Reviewer #1: No

Reviewer #2: No

Reviewer #3: No

---

## [Author Response · Author response to Decision Letter 0]

8 Oct 2024

Reviewer #1: The study titled ATM-deficient murine thymic T-cell lymphoblastic lymphomas are PTEN-deficient and require AKT signaling for survival by An et al delves into the genetic and signaling aspects of T-cell lymphoblastic lymphoma cells from ATM-deficient mice, a model mimicking human T-ALL. The authors show that most ATM-deficient thymic T-LBLs exhibit genetic alterations in Pten, leading to the loss of PTEN expression and increased AKT activation. Furthermore, they show that all ATM-deficient T-LBLs are sensitive to AKT inhibition, suggesting the potential therapeutic potential of AKT inhibitors in treating human T-ALL.

Although this study is conceptually interesting, the presented results are rather weak.

Furthermore, several questions arise while reading the manuscript and the authors may address them during the revision process:

- The first sentence of the abstract starts with the words “mice deficient in ATM…”. This is misleading as one immediately assumes that the experiments in this study are conducted in vivo. This is, however, not the case as the data in the manuscript are exclusively derived from cell culture-based experiments. In the Methods section, it is furthermore unclear, how the cells were extracted from mice to generate the analyzed cell lines.

RESPONSE: Although the bulk of the data came from lymphoma cells acquired shortly after harvesting the cells from mice, because some of the data are from cells after they have been propagated multiple times in vitro, I have adjusted some of the words to reflect that this was an in vitro study. I have also modified the methods section to reflect how the cells were obtained. 

- The figures are pixelated and difficult to decipher

RESPONSE: We have worked on getting this fixed

- The figure captions are somehow embedded in the text and are easily missed.

RESPONSE: The addition of the figure captions into the main text body of the manuscript was done according to the guidelines of the PLOS journal. 

- In Figure 6A, define non-malignant versus malignant cells

RESPONSE: Like in Petinot et al. 2000, we determined tumors by their significantly increased weight and size of the thymus compared to those of littermates without ATM deficiency. Necropsy was performed on all mice upon detection of rapid/shallow breathing or spontaneous death. Tumors frequently had an abnormal immunophenotype by expressing CD25/CD44 despite expressing CD4 and/or CD8. In addition, in contrast to T-LBLs, non-malignant thymocytes cannot self-propagate in our standard culture media. 

- In reference to Figure 6, would deleting Pten, e.g. using CRISPR-Cas9, in the PTEN-sufficient T-LBL cells mimic the phenotype observed in the PTEN-deficient cell lines?

RESPONSE: Are you referring to PTEN deletion in PTEN+ non-malignant ATMKO thymocytes or in PTEN+ ATMKO T-LBL cells? In PTEN+ non-malignant thymocytes, selective deletion of Pten using either Lck-cre or CD4-Cre models (Hagenbeek and Spits, 2008; Liu et al., 2010) have been described and do indeed promote T cell lymphomagenesis. Our ATMKO lymphoma model, contrary to the one aforementioned which involves genetically knocking out Pten, demonstrates that these lymphomas undergo a selective pressure to lose PTEN activity. In addition, most of the lymphomas generated from the CD4/Lck-Cre model are more developmentally mature (after pre-TCRB expression/the beta checkpoint), whereas some of our tumors possess a more developmentally early immunophenotype that resembles human pre-T cell ALL. In regards to PTEN+ T-LBLs, because these cells are still sensitive to the pan-AKT inhibitor, I don't expect a change in consequence if targeting Pten by CRISPR, at least in cell viability.

- What are the functional properties of the malignant T-LBL cells? Are they more proliferative?

RESPONSE: I can answer these questions in comparison to the non-malignant thymocytes. Whereas normal thymocytes are unable to proliferate in standard culture media, our T-LBL cells can propagate virtually indefinitely in vitro. In vivo, these T-LBLs proliferate to the point of causing mass effect in the mediastinum, ultimately leading to premature death of these mice.

- Would these cells be resistant to standard treatment of, e.g., T-ALL?

RESPONSE: The standard of care treatment for T-ALL is a multi-chemotherapy regimen such as HyperCVAD, which has a very high initial complete response rate. Thus, I would expect a very good anti-lymphoma response with these chemotherapeutics in vitro. The issue is that disease recurrence in vivo is common, and that response to salvage therapy is very poor.

- Are the T-LBL cells responsive to Everolimus treatment?

RESPONSE: We have some data indicating high sensitivity of our T-LBL cells to rapamycin but not everolimus. We did not include the data due to insufficient replicated data

Reviewer #2: The authors reported the frequent pTEN deficiency either by complete deletion, partial deletion or mutations of PTEN, inducing pAKT on Ser 473 and sensitivity to inhibitors of AKT signaling among the 12 ATMKO mice with T-LBL in this series.

Minor comments

1- The authors should explain how they select these 12 ATMKO T-LBL compared to their littermate of ATMKO mice and ATMKO with T LBL non selected for this study

RESPONSE: We included for analysis all of the 12 lines successfully adapted into in vitro culture

2-the p values for figure 6B should be given

RESPONSE: The p-values for Fig 6B were listed in the figure legend instead of in the manuscript itself, which have been fixed

Reviewer #3: In this manuscript by An et al., the authors confirm a previous discovery that ATMKO T-LBLs tend to exhibit Pten genetic alterations. The study characterized the mutations and found that most ATMKO T-LBLs have various genomic alterations in the Pten gene, resulting in the loss of functional PTEN protein. Consistently, all T-LBLs in the study demonstrated higher levels of phosphorylated AKT (pAKT), indicating the activation of AKT signaling. Importantly, this pAKT signaling appears to be crucial for the survival of the lymphomas, as indicated by the data on the AKT inhibitor MK-2206. The experiments were well-designed, and the results seem robust. This work thus nicely documents an interesting finding.

I have no criticism except that in Figures 1, 3, 4, and 5, the authors only present data from representative ATMKO T-LBLs. Considering there are only 12 of them and not much mechanistic insight was demonstrated in this work, I think that all the data should be included in this manuscript, at least in the supplementary materials.

RESPONSE: We have adjusted the figures to include all of the lines analyzed

---

## [Decision Letter · Decision Letter 1]

15 Oct 2024

ATM-deficient murine thymic T-cell lymphoblastic lymphomas are PTEN-deficient and require AKT signaling for survival

PONE-D-24-21090R1

Dear Dr. Hodes,

We’re pleased to inform you that your manuscript has been judged scientifically suitable for publication and will be formally accepted for publication once it meets all outstanding technical requirements.

Kind regards,

Alvaro Galli

Academic Editor

PLOS ONE

Additional Editor Comments (optional):

Reviewers' comments:

Reviewer's Responses to Questions

**Comments to the Author**

1. If the authors have adequately addressed your comments raised in a previous round of review and you feel that this manuscript is now acceptable for publication, you may indicate that here to bypass the “Comments to the Author” section, enter your conflict of interest statement in the “Confidential to Editor” section, and submit your "Accept" recommendation.

Reviewer #1: All comments have been addressed

Reviewer #2: All comments have been addressed

2. Is the manuscript technically sound, and do the data support the conclusions?

Reviewer #1: Yes

Reviewer #2: Yes

3. Has the statistical analysis been performed appropriately and rigorously? 

Reviewer #1: Yes

Reviewer #2: Yes

4. Have the authors made all data underlying the findings in their manuscript fully available?

Reviewer #1: Yes

Reviewer #2: Yes

5. Is the manuscript presented in an intelligible fashion and written in standard English?

Reviewer #1: Yes

Reviewer #2: Yes

6. Review Comments to the Author

Reviewer #1: Dear authors, thank you for addressing my remarks. Please pay attention to the figure quality, they still appear extremely blurry in the new version

Reviewer #2: the authors responded adequately to all the comments made and modified the manuscript as well as figures accordingly

7. PLOS authors have the option to publish the peer review history of their article (what does this mean?). If published, this will include your full peer review and any attached files.

Reviewer #1: No

Reviewer #2: No

---

## [Editor Report · Acceptance letter]

25 Nov 2024

PONE-D-24-21090R1 

PLOS ONE

Dear Dr. Hodes, 

I'm pleased to inform you that your manuscript has been deemed suitable for publication in PLOS ONE. Congratulations! Your manuscript is now being handed over to our production team.

Kind regards, 

on behalf of

Dr. Alvaro Galli 

Academic Editor

PLOS ONE